# The Biology of Prostaglandins and Their Role as a Target for Allergic Airway Disease Therapy

**DOI:** 10.3390/ijms21051851

**Published:** 2020-03-08

**Authors:** Kijeong Lee, Sang Hag Lee, Tae Hoon Kim

**Affiliations:** Department of Otorhinolaryngology-Head & Neck Surgery, College of Medicine, Korea University, Seoul 02841, Korea; peppermint_1111@hotmail.com (K.L.); sanghag@kumc.or.kr (S.H.L.)

**Keywords:** prostaglandins, allergy, asthma, allergic rhinitis, AERD, PGD_2_, PGE_2_

## Abstract

Prostaglandins (PGs) are a family of lipid compounds that are derived from arachidonic acid via the cyclooxygenase pathway, and consist of PGD_2_, PGI_2_, PGE_2_, PGF_2_, and thromboxane B_2_. PGs signal through G-protein coupled receptors, and individual PGs affect allergic inflammation through different mechanisms according to the receptors with which they are associated. In this review article, we have focused on the metabolism of the cyclooxygenase pathway, and the distinct biological effect of each PG type on various cell types involved in allergic airway diseases, including asthma, allergic rhinitis, nasal polyposis, and aspirin-exacerbated respiratory disease.

## 1. Introduction

Prostaglandins (PGs) are lipid mediators, generated from arachidonic acid (AA) metabolism via cyclooxygenases (COX). They were discovered in the 1930s as regulators of blood pressure and smooth muscle contraction [1]. The distribution of synthases and receptors for each PG is different in various cell types, and PGs are activated via either paracrine or autocrine signaling on the surface of each cell type [2]. PGs bridge the interactions between various immune-modulating cells, and are considered key players in regulating pro-inflammatory and anti-inflammatory responses [3]. The role of PGs in allergic disease was first introduced in an article published in 1969, which reported the release of PGE_2_ and PGF_2α_ during anaphylaxis in an animal model [4]. Since then, numerous studies have reported on the role of PGs in allergic diseases, and technologies have been developed to produce stable analogs and generate genetically engineered animal models [5].

Allergic reactions are characterized as immunoglobulin E (IgE)-dependent mechanisms, orchestrated by various cellular components and resulting in antigen-induced type 2 inflammation. Allergic diseases in the upper and lower airways include asthma, allergic rhinitis, and nasal polyposis. In airway allergic reactions, antigens that have penetrated epithelial layers are engulfed by dendritic cells (DCs), inducing them to mature and migrate to the lymph nodes, where they activate naïve T cells into effecter T helper 2 (Th2) cells [6]. Cytokines such as interleukin-4 (IL-4), IL-5, and IL-13 are released from Th2 cells or type 2 innate lymphoid (ILC2) cells. These cytokines, as well as epithelial-derived cytokines such as thymic stromal lymphopoietin (TSLP) and IL-33, activate the mast cells, eosinophils, and other structural cells, including smooth muscle cells and fibroblasts. These cells can also interact with each other to induce hyperreactivity [7]. Although aspirin-exacerbated respiratory disease (AERD) is not induced by the IgE-mediated hypersensitivity reaction, it is characterized by asthma, eosinophilic nasal polyposis, and nonsteroidal anti-inflammatory drug (NSAID) sensitivity, and it is a serious airway disease that affects both the upper and the lower airways [8]. AERD is dominantly regulated by lipid mediators such as PGs and leukotrienes (LTs) because the disease pathogenesis is characterized by alteration of AA metabolism [9,10].

Although PGs and their role in allergic reactions have been known about for a long time, the development of novel drugs targeting their signaling and metabolism is currently in progress, and clinical studies applying these to allergic airway diseases are being undertaken. In this review, we will discuss the potential of PGs to serve as therapeutic targets in allergic airway inflammation of the upper and lower airways. Based on an introduction of the pathways of PG generation, and the mechanisms of individual PGs in important cellular allergic reactions, we will present the regulation of PGs in clinical settings and the effects of pharmacological agents that modulate PG activation in various airway diseases such as asthma, allergic rhinitis, AERD, and nasal polyposis.

## 2. Biosynthesis of Prostaglandins

### 2.1. Phospholipase A_2_ and Arachidonic Acid

All PGs and LTs derived from AA are called “eicosanoids,” following the Greek word “eikosi,” meaning “20,” referring to the number of carbon atoms in AA. Free fatty acids, including AA, are obtained from the hydrolysis of fatty acids by phospholipase A_2_ (PLA_2_) at the sn-2 position of membrane phospholipids [11]. PLA_2_ enzymes are classified into six major groups: secretory PLA_2_ (sPLA_2_), cytosolic PLA_2_ (cPLA_2_), Ca^2+^ dependent PLA_2_ (iPLA_2_), platelet-activating factor acetylhydrolases (PAF-AH), lysosomal PLA_2_, and adipose-specific PLA_2_ [12]. In addition, based on their catalytic mechanisms, functions, and structures, they are further categorized into 16 groups. Among them, groups IIA/V/X of the sPLA_2_ group, group IVA of cPLA_2_, and group VI of iPLA_2_ lead to AA production by hydrolyzing membrane phospholipids, which are then metabolized sequentially to produce PGs and LTs [13].

### 2.2. The Cyclooxygenase Pathway

PGs are derived from the COX pathway [14]. COX catalyzes the initial reaction in the generation of PGG_2_ by inserting two oxygen molecules into AA (Figure 1). Thereafter, sequential peroxidase reactions transform PGG_2_ into PGH_2_, the precursor for PGD_2_, PGE_2_, PGF_2α_, PGI_2_, and thromboxane A_2_ (TXA_2_) [14].

In humans, COX-1 and COX-2 comprise the functional COX enzymes, whereas COX-3, which is encoded by COX-1, does not have known function yet. The two homologous COX enzymes have distinct genetic locations and roles in inflammatory reactions. In humans, COX-1 and COX-2 are located on chromosomes 9 and 1, respectively [14]. The expression of COX-1 is constitutive in most tissues, where it participates in the synthesis of homeostatic PGs [15]. However, COX-2 expression is usually transient, and induced by cellular stress or inflammatory reactions, for example after stimulation by lipopolysaccharide (LPS) or the secretion of cytokines, such as interleukin (IL)-1, IL-2, and tumor necrosis factor (TNF)-α. COX-2 has been proposed to have a central role in the production of a wide range of PGs during the inflammatory response [15].

### 2.3. Generation of Individual PGs and Their Receptors

PGH_2_ is believed to be converted immediately to its downstream product rather than accumulating in cells. Except for PGF_2α_, PGH_2_ metabolites such as PGD_2_, PGE_2_, PGI_2_, and TXA_2_ require specific synthases (Figure 1) [14]. All PGs elicit their biological effects by activating cell surface G protein-coupled receptors (GPCRs)—seven transmembrane spanning receptors [16]. PG receptors are classified into nine subfamilies (DP1, DP2, EP1–4, FP, IP, and TP) according to their affinities for individual PGs (Table 1) [17].

Two groups of synthases are involved in PGD_2_ production—hematopoietic-PGD_2_ synthases (H-PGDS) and lipocalin-PGD_2_ synthases (L-PGDS) [18]—of which only H-PGDS are responsible for PGD_2_ production by mast cells or other hematopoietic cells, such as eosinophils [19,20]. In humans, H-PGDS are abundantly expressed in the lungs, adipose tissue, placenta (at the tissue level), mast cells, T cells, dendritic cells, and megakaryocytes (at the cellular level) [18]. PGD_2_ signals through two receptors: DP1 and DP2 (also known as CRTH2). In the airways, DP1 is expressed in the nasal mucous-secreting goblet cells, nasal serous glands, vascular endothelium, DCs, T cells, basophils, and eosinophils [21]. DP2 expression has been observed in both hematopoietic cells such as T cells, ILC2s, mast cells, basophils, and eosinophils, as well as in structural cells such as epithelial cells and smooth muscle cells [22]. Stimulation of DP1 increases the intracellular cyclic adenosine 3′,5′-monophosphate (cAMP) level, which is believed to inhibit target cell functions and suppress innate immune functions, for example the generation of inflammatory mediators. Conversely, PGD_2_ signaling through DP2 decreases the cAMP level [14,23]. In addition to these functions PGD_2_ can activate the thromboxane receptor (TP), even at low concentrations [24].

Synthesis of PGE_2_ involves three distinct PGH_2_-metabolizing enzymes: microsomal PGE synthase (mPGES)-1 and -2, and cytosolic PGE synthase (cPGES) [14]. mPGES-1 is glutathione-dependent, with an inducible expression, and it preferentially catalyzes PGH_2_ produced by COX-2. In contrast, mPGES-2 utilizes PGE_2_ derived from either COX-1 or COX-2, and it shows constitutive expression in various cell types, including human embryonic kidney cells (HEK293), human colon adenocarcinoma cells (HCA-7) and human lung epithelial cells (BEAS-2B) [25]. cPGES is also expressed constitutively, and it generates PGE_2_ using COX-1, rather than COX-2. Four distinct receptors, termed EP receptors 1–4, are activated by PGE_2_, and are present in various organs associated with allergic reactions, including the lungs [26]. Stimulation of the EP1 receptor causes increase in inositol triphosphate (IP3) and diacylglycerol, followed by increase in intracellular calcium concentrations. Signaling through the EP2 and EP4 receptors is known to increase the intracellular cAMP level, whereas EP3 activation results in decreased cAMP synthesis in cells [14].

The formation of PGF_2α_ is catalyzed by PGF synthase (PGFS) via two pathways, utilizing NADPH and PGD_2_, respectively [27]. PGFS has been identified in lung and peripheral blood lymphocytes, suggesting their role in inducing airway allergic reactions [27]. A single receptor, termed FP, mediates PGF_2α_ signaling. FP has high affinity for PGF_2_, and its stimulation results in an increase in intracellular calcium concentrations [16].

PGI_2_ is generated from PGH_2_ by PGI synthase, which is highly expressed in the lungs, smooth muscle, heart, kidneys, and ovaries [28]. PGI_2_ activates adenylate cyclase and increases intracellular cAMP levels by stimulating its receptor, known as IP [16], which is strongly expressed in the thymus, lungs, heart, spleen, and neurons [28].

TXA_2_ accounts for the majority of AA metabolites, but it is unstable in nature, with a short half-life of 30 s [26]. TXA_2_ is hydrolyzed to thromboxane B_2_ (TXB_2_) in the absence of an enzyme. The synthase that catalyzes the formation of TXA_2_ from PHG_2_ is named thromboxane synthase, which shows strong expression in the lungs, kidneys, liver, monocytes, and megakaryocytes [29]. TXA_2_ is predominantly produced by cells such as platelets, neutrophils, monocytes, macrophages, and lung parenchyma cells [30]. TXA_2_ signals through its receptor TP, which has two isoforms, TPα and TPβ. Although both of these isoforms lead to phospholipase C activation, calcium release, and protein kinase C activation, they have opposing functions when coupled to a Gq protein: TPα activates adenylate cyclase, followed by increased cAMP levels and the induction of cAMP-dependent intracellular signaling pathways, while TPβ inhibits them [31].

## 3. The Role of Prostaglandins in Various Cell Types Involved in Allergic Reactions

Allergic respiratory reactions occur because of interactions between various cells and cytokines [32]. When an allergen reaches the respiratory mucosa, epithelial cells and ILC2 secrete cytokines, while DCs present allergens, resulting in Th2 differentiation of naïve T cells and IgE production from B cells [33,34]. Eosinophilic inflammation caused by intercellular reactions leads to epithelial mucus hypersecretion, bronchoconstriction, and airway remodeling. PGs affect all stages of this reaction [35] (Figure 2).

### 3.1. Epithelial Cells

In human nasal polyps, stronger expression of H-PGDS and PGD_2_ was observed than in normal nasal mucosa. Moreover, PGD_2_ in airway epithelial cells induced increased MUC5B expression and mucin hypersecretion by the DP1 receptor [36,37]. A recent study using primary bronchial epithelial cell reported that PGD_2_ promoted epithelial cell migration as well as goblet cell hyperplasia, which were attenuated by DP2-selective antagonist [38].

Human bronchial epithelial cells treated with IL-13 showed increased airway mucin production, with increased expression of the mucin gene MUC5AC after PGE_2_ stimulation via the EP4 receptor [39]. Moreover, epithelial cells derived from the nasal polyps of patients with aspirin sensitivity showed dysregulated AA metabolism, which abnormally reduced the spontaneous production of PGE_2_, suggesting a mechanism of chronic inflammation in these patients [40]. PGE_2_ derived from epithelial cells also affected cells such as DCs and smooth muscle cells. Activation of the EP4 receptor by PGE_2_ released from airway epithelial cells limits DC activation and reduces the pro-inflammatory properties of DCs [41]. In addition, epithelia-derived PGE_2_ generated in response to bradykinin stimulation reduces airway smooth muscle contraction [42].

### 3.2. Dendritic Cells

The maturation and migration of DCs is affected by PGs (Figure 3) [2]. In the presence of PGE_2_, CCL7 ligands triggered DC migration through the activation of signaling pathways, including cAMP-dependent PKA and rho kinase [43]. Interestingly, the presence of PGE_2_ was found to be effective only at initiation, and not termination, of DC maturation. In addition, modulation of DC migration by PGE_2_ reportedly requires a signal in the form of stimulation of either EP2, EP4, or both [44,45]. In contrast, PGD_2_ and PGI_2_ reduce DC maturation and migration. The inhibitory effect of PGD_2_ administration was observed in Balb/c mice exposed to fluorescein isothiocyanate (FITC)-OVA inhalation as a decreased migration of FITC+ DC to draining lymph nodes. This effect was also mediated by a DP1 agonist, but not by a DP2 agonist, suggesting that DP1 is responsible for modulating DC migration [46]. Another study showed that DP1 activation in DCs induced FOXP3 (+), CD4 (+), and regulatory T (Treg) cells, but DP1 depletion in DCs enhanced Th2 response in the airways, supporting this hypothesis [47]. In a murine model of asthma, inhalation of a stable PGI_2_ analog, iloprost, reduced the maturation and migration of lung DCs, thereby reducing the allergen-specific Th2 response in mediastinal lymph nodes [48]. In addition, iloprost treatment of DCs decreased differentiation of naïve T cells into effector Th2 cells [49].

PGE_2_ also has an important role in DC–T cell interactions as well as T cell differentiation (Figure 3). For Th1/Th2 balance, the ratio between PGE_2_ and IL-12 from DCs is important, suggesting that PGE_2_ induces Th2-skewing responses by inhibiting IL-12 production [50,51,52]. Reduction of IL-12 by PGE_2_ also inhibits interferon (IFN)-γ release from T cells and natural killer cells [51,53]. A recent study revealed a new mechanism for activation of Th2 immune response mediated by DCs, wherein allergen-induced PGE_2_ secretion by DCs resulted in autocrine OX40 ligand upregulation [54]. In a study on murine bone marrow-derived DCs (BMDCs), PGE_2_ inhibited TNF-α and IL-6 secretion by inducing IL-10 [55,56,57]. However, in the presence of TNF-α, PGE_2_ stimulation of human BMDCs increased IL-12 production, inducing a Th1 response [58,59]. In addition, BMDCs stimulated by PGE_2_ promoted IL-23 production, and induced Th17 differentiation [60]. In a recently published study of an OVA-induced asthma model, treatment of DCs with a PGI_2_ analogue, iloprost, induced differentiation of naïve T cells to Treg cells, leading to immune tolerance [61].

### 3.3. T Cells

PGD_2_ production was found to occur preferentially in antigen-stimulated human Th2 cells, but not in Th1 cells (Figure 3) [26]. PGD_2_ inhibits IFN-γ secretion by activating DP1, and stimulates Th2 cytokine expression through DP2 (CRTH2) activation [62]. A more recent study showed that the IL-5 positive effector T cell population expressing the CRTH2 receptor, was the pro-inflammatory Th2 cell population and induced allergic inflammation [63]. Moreover, mast cell-derived PGD_2_ activated type-2 CD8 (+) T cells via the CRTH2 receptor, inducing migration and cytokine (IL-5 and IL-13) production in severe eosinophilic asthma [64]. Studies supporting these findings have shown that a larger population of CRTH2 (+) T cells was identified from the peripheral blood of patients with severe asthma, and allergen-specific Th2 cells could be identified by stable co-expression of CRTH2 [65,66,67].

PGE_2_ has been proposed to induce Th2 responses by attenuating the production of IL-2 and IFN-γ by Th1 cells without enhancing the release of IL-4 by Th2 cells [26,68]. In an in vitro study, PGE_2_ stimulation of CD4 (+) T cells downregulated not only Th1 cytokines such as IFN-γ and TNF-α, but also IL-4. Nevertheless, differentiation skewed toward Th2, because of a much stronger inhibition of Th1 cytokine production [69]. However, studies regarding individual PGE_2_ receptors have shown conflicting results. Increased cAMP signaling via PGE_2_–EP2/EP4 in naïve T cells promoted the upregulation of IL-12Rβ2 and IFN-γ R1, thereby inducing Th1 differentiation [70]. The endogenous PGE_2_–EP2 axis was shown to have a role in controlling allergen sensitization and reducing theTh2 immune response in a murine model of asthma [71]. Furthermore, recent studies have shown that endogenous PGE_2_ coupled with the EP2–EP4 receptor induces the activation of pathogenic Th17 cells in both murine and human T cells [72,73]. Another study showed that PGE_2_ stimulation of T cells from the peripheral blood of patients with allergic rhinitis caused them to differentiate into Treg cells via EP4 receptor activation [74]. Although conflicting results have been reported on whether PGE_2_ induces differentiation of naïve T cells to Th1 or Th2, there is a consensus on the effect of PGE_2_ in Treg cell-induced inhibition of effector T cells [75].

### 3.4. B Cells

Most studies on the role of PGs in B cells have investigated the effect of PGE_2_ on the development of B cells and their production of IgE. However, studies on the role of PGE_2_ on IgE production have shown conflicting results. In the most recent study, B cells from EP2-deficient mice showed impaired IgE production and airway inflammation after an OVA challenge, suggesting the role of PGE_2_–EP2 signaling in promoting IgE production in allergic inflammation [76]. In support of this finding, PGE_2_ was shown to promote immunoglobulin class switching to IgE in B cells in the presence of IL-4 and LPS [77]. In contrast, PGE_2_ was shown to reduce MHC-II expression in B cells and suppress IgE production, which was mediated by either EP2 or EP4 [78]. Another study on B cell responses in an OVA-induced asthma murine model also showed that EP2 deficiency resulted in elevation of serum IgE level [71]. Taken together, although the exact effect of PGE_2_ is unclear, these studies emphasize the role of PGE_2_ in modulating IgE production.

### 3.5. Type 2 Innate Lymphoid Cells

The relationship between ILC2 cell function and PGs in allergic disease was first reported in a study showing that PGD_2_ released from mast cells induced IL-13 production by ILC2 cells, via DP1 or DP2 receptors (Figure 4) [79]. ILC2 cells obtained from the peripheral blood of patients with allergic rhinitis induced IL-5 production after stimulation by PGD_2_ [80]. Furthermore, activation of DP2 receptor on human ILC2 cells induced their migration, production of type 2 cytokines, and increased expression of receptors for IL-33 and IL-25 [81]. ILC2 accumulation, induced by IL-33, was not observed in a DP2-deficient murine model [82]. In the sputum of patients with asthma, a DP2-positive ILC2 population, in addition to IL-5- or IL-13–positive ILC2s, was observed to be significantly higher after allergen challenge than in the controls [83]. In addition, the number of ILC2s after allergen challenge was elevated in the bronchoalveolar lavage (BAL) fluid and reduced in the blood. PGD_2_ levels in the BAL fluid showed a significant correlation with a decrease in ILC2 in the blood, indicating that PGD_2_ is essential for ILC2 migration [84]. In patients with AERD, recruitment of ILC2 to the nasal mucosa was observed upon treatment with a COX-1 inhibitor, with a corresponding decrease in ILC2 numbers in the blood, supporting the correlation with enhanced PGD_2_ production [85]. Further supporting these findings, ILC2s isolated from human blood that were treated with the PGD_2_ antagonist, fevipiprant, showed reduced aggregation and diminished cytokine production [86]. Moreover, human ILC2 cells treated with the H-PGDS enzyme inhibitor, KMN698, showed a diminished release of IL-5 and IL-13 [87].

Among four receptors for PGE_2_, only the EP2 and EP4 receptors showed strong expression in single-cell transcriptome sequencing of human tonsillar ILC2 cells [88]. In the same study, PGE_2_ attenuated ILC2 function by inhibiting GATA-3 expression, and decreasing the release of Th2 cytokines, including IL-5 and IL-13. Another study revealed that PGE_2_–EP4–cAMP signaling inhibited IL-33-induced Th2 cytokine production in pulmonary ILC2 cells by suppressing GATA3 and ST2 expression [89]. Similarly, PGI_2_ inhibited the expression of IL-5 and IL-13 in IL-33-stimulated ILC2 cells from the bone marrow in a mouse model challenged intranasally with *Alternaria alternata* [90].

### 3.6. Eosinophils

PG receptors expressed on human eosinophils include DP1 and DP2 for PGD_2_, EP2 and EP4 for PGE_2_, and IP for PGI_2_ (Figure 5) [91].

Both DP1 and DP2 are present on the surface of eosinophils [92]. In addition to PGD_2-_induced stimulation of these receptors, the major metabolite of PGD_2_, 15-deoxy-Δ12,14-PGJ_2_, activates the peroxisome proliferator-activated receptor-γ on eosinophils [93]. PGD_2_ is known to modulate eosinophil migration via the DP2 receptor [94,95]. PGD_2_-DP2 signaling also enhances Ca^2+^ morphologic changes and degranulation of eosinophils in allergic inflammatory responses [94]. A recent study revealed that blocking the PGE_2_-DP2 pathway inhibits migration of eosinophils toward mast cells—the initial stage in an allergic reaction [96]. In addition, by mediating its effect through DP2, PGD_2_ also enhances the function of other chemoattractants, such as eotaxin, complement factor C5a, or 5-oxo-6,8,11,14-eicosatetraenoic acid (5-oxo-ETE) on eosinophils [97,98]. In contrast, eotaxin and 5-oxo-ETE diminished eosinophil migration toward PGD_2_, and their effect was decreased in the presence of blood or plasma, while no effect was identified for PGD_2_, suggesting that it acts as the initial chemoattractant, while eotaxin is the endpoint chemoattractant [98]. The DP1 receptor on eosinophils inhibits eosinophil apoptosis and cooperates with DP2 receptors to induce LTC_4_ synthesis, eosinophil mobilization, and pro-inflammatory signaling [92,94,99,100]. In eosinophils obtained from the peripheral blood of patients with AERD, high expression of H-PGDS was identified, and inhibitory action of the H-PGDS enzyme suppressed the release of PGD_2_ [101]. This suggests that eosinophil activation could occur via autocrine signaling in patients with AERD and allergic inflammation [20].

In eosinophils, PGE_2_ decreases the production of the eosinophil cationic protein and the aggregation of eosinophils [102,103]. EP2 receptor activation in eosinophils attenuates eosinophil trafficking and PGE_2_–EP4 signaling, and decreases migration and degranulation of eosinophils [104,105]. PGE_2_ also inhibits the spontaneous apoptosis of eosinophils in human peripheral blood and suppresses eosinophil–endothelial cell interactions, including adhesion and transmigration, by altering β_2_ integrin and L-selectin function [106,107]. Eosinophils obtained from human peripheral blood showed reduced production of LTB_4_ and cysteinyl LTs (cysLTs) when treated with a low dose of PGE_2_ after lysine aspirin stimulation, and this effect was mediated by EP2 receptor activation [108].

PGI_2_ has similar properties to PGE_2_, as both act as immune suppressors in eosinophils through the IP receptor, and both modulate intracellular cAMP [91]. In guinea pigs, both PGI_2_ and a PGI_2_ analogue, iloprost, inhibit bone marrow eosinophil trafficking [109]. In addition, endothelial-derived PGI_2_ negatively modulates eosinophil–endothelial interactions by inhibiting eosinophil adhesion and transendothelial migration [110].

### 3.7. Mast Cells

Mast cells are an important source of endogenous eicosanoids, and PGD_2_ is the predominantly released mediator, with cysLTs. Production of PGD_2_ from mast cells isolated from human lungs depends on COX-1 and H-PGDS, but not L-PGDS [111]. In the nasal mucosa of patients with allergic rhinitis, the number of mast cells with H-PGDS expression was high, while that with L-PGDS expression was low [21]. In humans, there was a linear correlation between PGD_2_ generation and histamine secretion after IgE-dependent activation of human mast cells [19]. A recent study, using nasal polyp tissue from patients with chronic rhinosinusitis and AERD, showed that TSLP induces the production of PGD_2_ by mast cells [112]. Another recent study showed intracellular expression of DP2 receptors in mast cells isolated from nasal polyps [113]. However, the majority of mast cells (either cell-line-derived human mast cells or cells harvested from the nasal cavity) did not express the PGD_2_ receptors, and did not respond to a DP2 agonist or antagonist [114].

Human mast cells also express EP2, EP3, and EP4 receptors for PGE_2_ signaling, and PGE_2_ stimulation coupled with EP2 activation inhibited mast cell degranulation, whereas EP3 receptor activation enhanced mast cell mediator release [115,116,117].

### 3.8. Smooth Muscle Cells

The role of PGs in modulating contraction of airway smooth muscles is widely accepted [39]. Four distinct receptors (PGE_2_, EP2, EP3, and EP4) have been detected on human airway smooth muscle cells [118]. Inhalation of PGE_2_ diminished bronchoconstriction in a methacholine airway test after an allergen challenge, and decreased bronchoconstriction induced by exercise or aspirin [119]. In humans, the EP4 receptor appears to mediate this effect, as a selective EP4 agonist could reverse the histamine-induced contraction of airway smooth muscle cells [120]. In other species, including mice, guinea pigs, and monkeys, PGE_2_-mediated regulation of smooth muscle contraction was regulated by EP2 [121]. Conversely, the activation of PGE_2_ is known to induce bronchial contraction, emphasizing the importance of receptor selectivity. In a recent study, bronchoconstriction caused by PGE_2_ and other PGs involved TP receptor activation pathways, and the powerful bronchoprotective effect of PGE_2_ was generated through mast cell-mediated bronchoconstriction via activation of the EP2 receptor, while EP4 stimulation had less efficacy than a long-acting β-receptor agonist [121]. In addition to muscle tone regulation in airway cultured smooth muscle cells, PGE_2_ limits the release of the granulocyte macrophage-colony stimulating factor, an important mediator of allergic inflammation and eosinophil survival [122]. Moreover, PGE_2_-induced stimulation of the EP2 or EP4 receptors inhibits migration of smooth muscle cells of the airways through the cAMP/PKA pathway, suggesting a potential pharmacological role of PGE_2_ in preventing airway remodeling in asthma [123].

In contrast to PGE_2_, PGD_2_ acts as a bronchoconstrictor and has a potent effect on muscle contraction, mediated via the TP receptor [24]. In humans, the increase in nasal resistance induced by PGD_2_ nasal instillation was greater than that of histamine (10-fold) or bradykinin (100-fold) [24]. A recent in vitro study showed that inhibition of the PGD_2_-DP2 pathway reduced the migration of airway smooth muscle cells, suggesting a role for DP2 antagonists in airway remodeling [124].

PGF_2α_ inhalation in healthy and asthmatic subjects caused bronchoconstriction in a dose-dependent manner; however, the sensitivity was approximately 150 times greater in asthmatic patients than in healthy controls [125]. In healthy controls, the responsive dose varied widely across individuals, but females generally had a smaller contractile response to PGF_2α_ than males [125].

TXA_2_ is also reported to be a potent bronchoconstrictor. In asthmatic patients, bronchoconstriction induced by antigen inhalation was shown to involve high concentrations of TXB_2_, which is a stable metabolite of TXA_2_, suggesting the possible role of TXA_2_ in airway remodeling or hyper-responsiveness [126]. This effect was mediated through c-Jun N-terminal kinase–mitogen-activated protein kinase signaling, and increased the extracellular calcium influx after stimulation of the TP receptor [127].

### 3.9. Fibroblasts (in Nasal Polyps)

The role of PGs in fibroblasts has also been studied, as formation of eosinophilic nasal polyps is a typical phenotype of patients with dysregulated AA metabolism.

Nasal polyp-derived fibroblasts showed increased production of vascular endothelial growth factor after PGE_2_ stimulation via EP4 receptor activation, indicating that PGE_2_ has a role in increased micro-vascular permeability during tissue remodeling in nasal polyps [128]. In a study using cultured fibroblasts from healthy nasal mucosa and nasal polyps from patients with aspirin-tolerant and -sensitive asthma, PGE_2_ concentration was lower in patients with both aspirin-tolerant and -sensitive nasal polyps after IL-1β stimulation, than in the controls [129]. In addition, increased expression of the EP2 receptor in fibroblasts after IL-1β stimulation was identified only in normal mucosa, but not in nasal polyps from either aspirin-tolerant or -sensitive patients [129]. Another study suggested that decreased expression of the EP2 receptor in fibroblasts from nasal polyps of patients with AERD resulted in resistance to the anti-proliferative effects of PGE_2_ [130].

PGD_2_ stimulation also induced the release of vascular endothelial growth factor, but this release occurred via the DP1 receptor, not the DP2 receptor [131].

## 4. Clinical Studies of Prostaglandins in Allergic Airway Diseases

The expression and role of PGs in allergic airway diseases have been studied for a long time. Recently, clinical phase II and III studies were conducted to evaluate the therapeutic effects of PGs, especially for asthma and allergic rhinitis (Table 2). This section briefly introduces the role of PGs in each disease (asthma, allergic rhinitis, AERD, and nasal polyps) in clinical settings, and provides an overview of the clinical studies that have used pharmacologic agents targeting PGs.

### 4.1. Asthma

Numerous studies have focused on the clinical relevance of prostaglandins in asthma. In a study that examined concentrations of PG mediators from BAL fluid, both PGD_2_ and PGF2α showed 12- and 20-fold higher levels in asthmatic patients than nonallergic controls, respectively [149]. Another study reported that concentrations of PGD_2_, TXB_2_, and PGI_2_ metabolites was much higher (up to 208-fold) in BAL fluid 5 min after an allergen challenge in allergic asthmatic patients compared to the controls [150]. In addition, epithelial cells obtained from severely asthmatic patients showed higher levels of H-PGDS compared to controls, correlating with PGD_2_ levels. Furthermore, BAL fluid collected from severely asthmatic patients showed stronger expression of DP2 receptors than that from mild asthmatics or heathy controls; however, there was no difference in DP1 receptor expression among groups [151]. As for PGE_2_ levels, studies regarding their correlation with eosinophils reported that a lower sputum PGE_2_ level is associated with eosinophilia in asthmatic patients [152].

To date, numerous clinical trials have shown that DP2 antagonists are effective for patients with asthma (Figure 1) [153,154,155]. Fevipiprant (QAW039), an oral DP2 antagonist, improved pulmonary function in asthmatic patients with severely impaired lung functions and patients with uncontrolled asthma using an inhaled corticosteroid (ICS) [132,133,156]. In a single-center randomized controlled trial, decreased sputum eosinophil count was observed in patients with moderate to severe asthma after treatment with fevipiprant [134]. Indeed, to date, fevipiprant is recognized as the most potent pharmacologic agent targeting PG receptors [157,158]. All phase II studies have agreed on the safety and tolerability of fevipiprant [159]. Several phase III trials are ongoing to determine its potential to prevent asthma exacerbation (LUSTER1 & LUSTER2), and its safety in moderate uncontrolled asthma in ICS-treated asthmatic patients (ZEAL1 & ZEAL2) [160,161,162,163,164]. Another DP2 antagonist, ARRY-502, improved one second-forced expiratory volume (FEV_1_) results as well as symptom-related quality of life of patients with asthma relative to control patients. This effect was greater in patients with elevated Th2-associated biomarkers [135]. Treatment with AZD1981, a DP2 receptor antagonist, showed improvements in the asthma control questionnaire score as well as FEV_1_ in patients with allergic asthma after four weeks [136]. OC000459 was also found to be safe and effective in inhibiting the DP2 receptor, showing significant improvements in pulmonary function in patients with eosinophil-dominant allergic asthma [137]. Other DP2 selective antagonists that were effective from phase II trials include GB001, BI671800, and setipiprant (ACT-129968) [139,140,141,165].

Recently, a phase I clinical trial for evaluating the efficacy of ZL-2010, an H-PGDS inhibitor, in asthma treatment has been completed, but the results are not yet published (Figure 1) [166].

In patients with asthma, PGE_2_ levels in sputum were negatively correlated with the eosinophil count, and inhaled PGE_2_ was effective in preventing pulmonary hyper-reactivity after an allergen challenge [167,168], but inhaled PGE_2_ did not alter baseline pulmonary functions without this challenge [168]. For investigating the effects of oral PGE_2_ administration, a synthetic PGE_2_ analog named misoprostol was used, because of the short half-life of PGE_2_ (Figure 1) [169]. However, no protective effect of misoprostol on pulmonary function, asthma symptom scores, or serum eosinophil count was identified in patients with aspirin-sensitive asthma, suggesting that oral administration could not deliver a sufficient concentration to the pulmonary system for pharmacologic activation [169].

Although PGF_2α_ has not been studied as much as PGD_2_ or PGE_2_, studies of its effect on allergic reactions have shown that PGF_2α_ inhalation decreases specific airway conductance in a dose-related manner, and induces bronchoconstriction, which can shorten the recovery by PGE_2_ [125,170]. In addition, inhaled PGF_2α_ decreased the exhaled nitric oxide (NO) level, although the clinical implications of this finding have not yet been identified [171].

The effects of allergic response treatment with PGI_2_ have been extensively studied; however, to our knowledge the most recent article was published in 1991 [172]. This might be because of the unstable nature of PGI_2_ that catalyzes within 3 to 5 min, and the absence of a stable PGI_2_ analog. Previous studies have shown that PGI_2_ inhalation did not exert a protective effect against bronchoconstriction after an allergen challenge, with no change in airway conductance [148,173]. Moreover, another study using an oral analog of PGI_2_ (OP-41483), showed no effect on FEV_1_ and airway responsiveness in patients with asthma (Figure 1) [172]. However, the converse result has also been reported—that PGI_2_ inhalation causes a dose-dependent decrease in FEV_1_ with no effect on airway conductance, suggesting narrowing of the airway due to mucosal blood engorgement via the vasodilatory effects of PGI_2_ [174].

As for TXA_2_, treatment of patients with asthma using a TP antagonist, seratrodast (AA-2414), resulted in improvement of symptom scores, bronchial responsiveness, and peak expiratory flow, and a reduction of inflammatory biomarkers in the epithelium of patients after four weeks [145,146].

### 4.2. Allergic Rhinitis

Clinical studies of allergic rhinitis have also suggested the importance of PGs in the disease. In nasal lavage fluid (NALF), the PGD_2_ level was higher in patients with allergic rhinitis compared to those with nonallergic rhinitis [175]. Nasal mucosa samples obtained from the turbinates of patients with allergic rhinitis revealed stronger DP2 expression in eosinophils, mast cells, T lymphocytes, and epithelial cells compared to the nonallergic nasal mucosa [176]. In patients with seasonal allergic rhinitis and sensitization to grass pollen, the PGE_2_ level in NALF decreased significantly during the pollen-exposure season, which was negatively correlated with nasal eosinophilia [177].

Although fewer than for asthma, several clinical trials of PG-targeting pharmacologic agents have been conducted for allergic rhinitis (Table 2). In a randomized, double-blind, placebo-controlled study, a DP2 antagonist (OC000459) was effective in reducing nasal and ocular symptoms in patients with allergic rhinitis caused by grass pollen [138]. Clinical trials of the dual DP2/TP receptor antagonist, ramatroban (BAY u3405), in patients with rhinitis showed conflicting results. In a study of patients with atrophic rhinitis, PGD_2_-induced nasal congestion was not relieved by the administration of a single 20-mg dose of ramatroban [178]. However, when administrated twice a day at a higher dose (150 mg) for four weeks, it effectively decreased eosinophil levels in the nasal lavage fluid and reduced mucosal swelling after house dust challenges in patients with perennial allergic rhinitis [143,144]. In a phase II clinical study, setipiprant showed a significant improvement in nasal symptoms, during the day and night, in patients with seasonal allergic rhinitis; however, in the phase III trial it did not show efficacy at the endpoint [142]. The DP1 antagonist, ONO-4053, also reduced nasal symptoms including rhinorrhea, sneezing, and nasal itching in patients with seasonal allergic rhinitis, without causing safety issues, in a phase II trial [147].

Although previous studies have shown that selective inhibition of H-PGDS (TFC-007 and TAS-204) is effective in reducing nasal blockage and eosinophil infiltration in animal models of allergic rhinitis, no studies have yet been conducted in humans to determine the role of H-PGDS inhibitors in allergic rhinitis [179,180].

### 4.3. Aspirin-Exacerbated Respiratory Disease 

AERD is a nonallergic disease caused by dysregulation of eicosanoids and characterized by eosinophilia, Th2 dominant cytokines, and a marked increase of CysLTs [181]. In AERD patients, COX-2 expression and PGE_2_ level are significantly reduced, because the *mPGES-1* gene is functionally coupled to COX-2. However, there was no change in the expression of COX-1. Conversely, the 5-lipoxygenase (5-LO) pathway, which metabolizes AA into LTs, is markedly upregulated in these patients, resulting in an extremely high level of CysLTs, which activate mast cells and eosinophils [10,182]. Addition of nonsteroidal anti-inflammatory drugs, especially COX-1 inhibitors, forces AA metabolism towards LT synthesis and further decreases PGE_2_ production, resulting in the uncontrolled release of LTC_4_, LTD_4_, LTE_4_, and PGD_2_ by countering the suppressive effect of PGE_2_ on 5-LO [183,184,185].

Studies of the nasal mucosa or nasal polyps of patients with AERD have shown lower expression levels of the EP2 receptor than in healthy patients, or patients with aspirin tolerance, suggesting that the decreased expression of EP2 in AERD might have a synergistic effect on the anti-inflammatory and bronchoprotective capacity of the PGE_2_–EP2 axis [186,187]. Interference with PGE_2_ production by oral administration of the PGE_2_ analog misoprostol did not protect against the symptoms induced by NSAIDs in patients with AERD [188].

The importance of PGD_2_ in AERD pathogenesis has also been studied [189]. In patients with AERD, an aspirin challenge did not reduce the PGD_2_ concentration in BAL fluid, unlike in patients with asthma [190,191]. Moreover, frequent aspirin challenges increased PGD_2_ metabolites in the urine and serum of patients with AERD, possibly owing to the abnormal activation of mast cells [192]. A study evaluating the role of PGD_2_ in AERD showed that severe extra-pulmonary reactions are associated with PGD_2_ production, and high-dose aspirin therapy could be used to inhibit PGD_2_ generation and suppress inflammatory cell recruitment [189]. In patients with AERD, intravenous injection of PGI_2_ had no effect in preventing decreased airflow after aspirin stimulation [193].

### 4.4. Nasal Polyps

Few studies have investigated PGs in nasal polyps, whereas comparatively more have assessed their roles in asthma or allergic rhinitis. However, the clinical relevance of PGs has been well demonstrated, even disregarding their association with aspirin sensitivity [194].

PGE_2_ concentrations and COX-2 mRNA expression were lower in nasal polyp tissues than in nasal mucosas, either from healthy controls or from patients with chronic rhinosinusitis without nasal polyps. Higher production of LT metabolites, such as LTC_4_, LTD_4_, and LTE_4_ was correlated with lower PGE_2_ concentrations [195]. Another study revealed a decreased level of PGD_2_ and its metabolites in the serum of patients with chronic rhinosinusitis [196]. Supporting this finding, the mRNA expression of PGDS and PGES was higher and lower, respectively, in nasal polyps than in the normal uncinate process, and a negative correlation was observed between the PGDS and PGES levels [197]. Moreover, the high PGDS and low PGES levels were correlated with increased eosinophil infiltration, as well as disease severity, suggesting that PGs might have a role in eosinophilic inflammation in nasal polyps [197].

Regarding the expression of individual PGE_2_ receptors in nasal polyps, the EP2 receptor showed a lower expression in immunohistochemistry of nasal biopsies from patients with both aspirin-sensitive and -tolerant chronic rhinosinusitis than controls. However, the expression was lowest in aspirin-sensitive patients [198,199]. Moreover, increased EP1 receptor expression was associated with the eosinophilic phenotype in patients with nasal polyps without aspirin intolerance [194]. In a recent study, exposure to cigarette smoking was related with diminished expression of EP2 and EP4 receptors in nasal polyps of patients with chronic rhinosinusitis, which might explain the severe inflammation caused by smoking [200]. In a study of PGD_2_ receptors in nasal polyp tissue, higher DP1 and lower DP2 receptor expression was identified relative to normal nasal mucosa, and both these findings correlated with increased levels of PGDS and eotaxin [201].

Despite the promising importance of PGs in nasal polyps reported by previous studies, to our knowledge, clinical trials for nasal polyps with pharmacologic agents modulating PGs have not yet been conducted, and future studies are required to provide clinical evidence.

## 5. Conclusions

In this review, we described the biosynthesis of PGs and their roles in various cell types relating to allergic airway inflammation. We summarized up-to-date information about the clinical relevance of PGs and clinical trials regulating their activation in each allergic airway disease. Owing to their short half-life, researchers have faced difficulties in conducting in vivo studies and clinical trials of PGs. However, with the development of stable PG analogs, their promising role in regulating allergic reactions, and the possibility for future therapeutic agents based on their biochemistry has been demonstrated by both animal and human studies. Nevertheless, little is known about their role in modulating the pathogenesis of nasal polyps, or the effect of PG synthase inhibitors on allergic inflammation. Future studies should focus on methods for maintaining the biological activity of PGs, and the development of pharmacological agents targeting PG synthase to assist in modulating allergic airway diseases, such as nasal polyps.

## Figures and Tables

**Figure 1 ijms-21-01851-f001:**
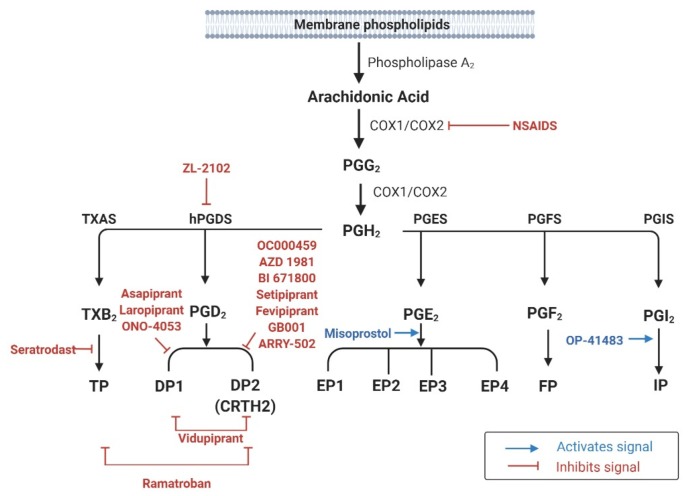
Prostaglandin biosynthesis pathways and pharmacologic agents used in clinical trials for human respiratory allergic diseases.

**Figure 2 ijms-21-01851-f002:**
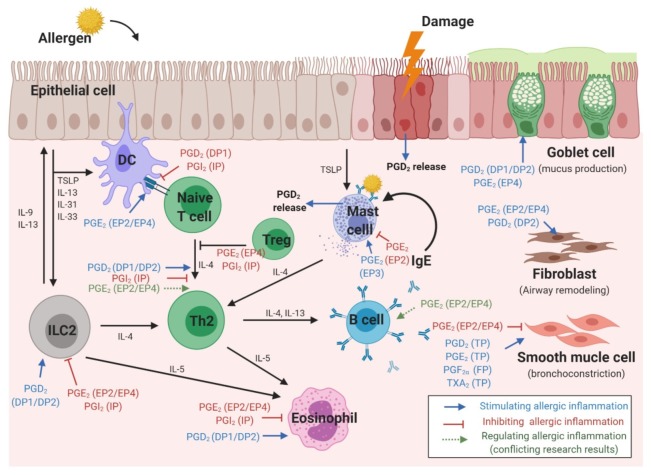
Stimulatory and inhibitory effects of prostaglandins, and their receptors, in different cell types involved in the pathophysiology of respiratory allergies.

**Figure 3 ijms-21-01851-f003:**
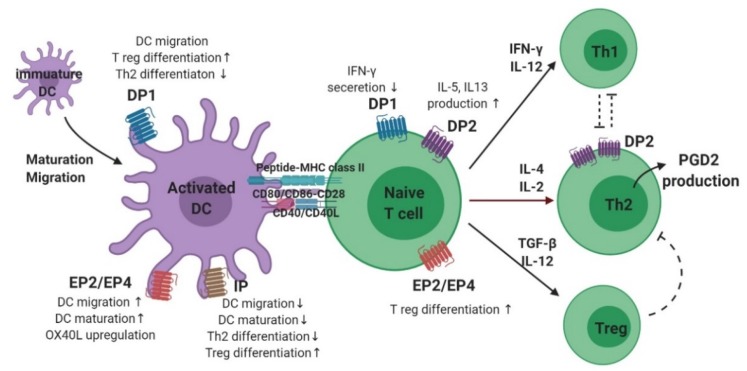
The role of prostaglandins and their receptors in dendritic cell–T cell interaction and Th2 differentiation.

**Figure 4 ijms-21-01851-f004:**
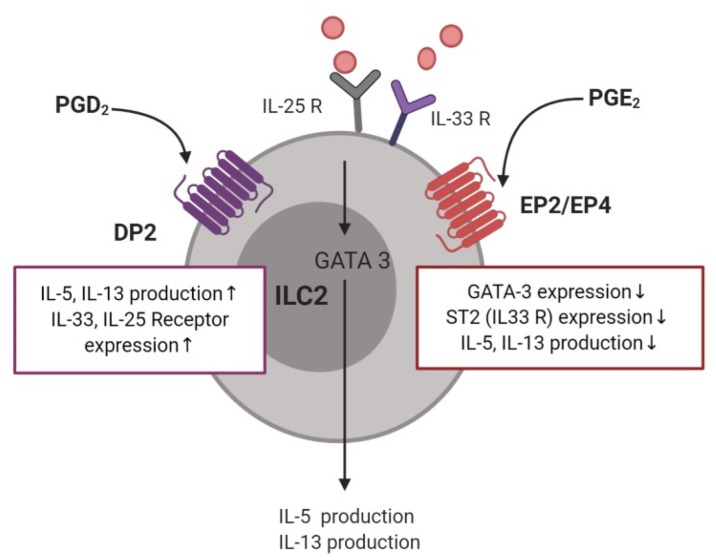
The role of prostaglandins and their receptors in type 2 innate lymphoid cells.

**Figure 5 ijms-21-01851-f005:**
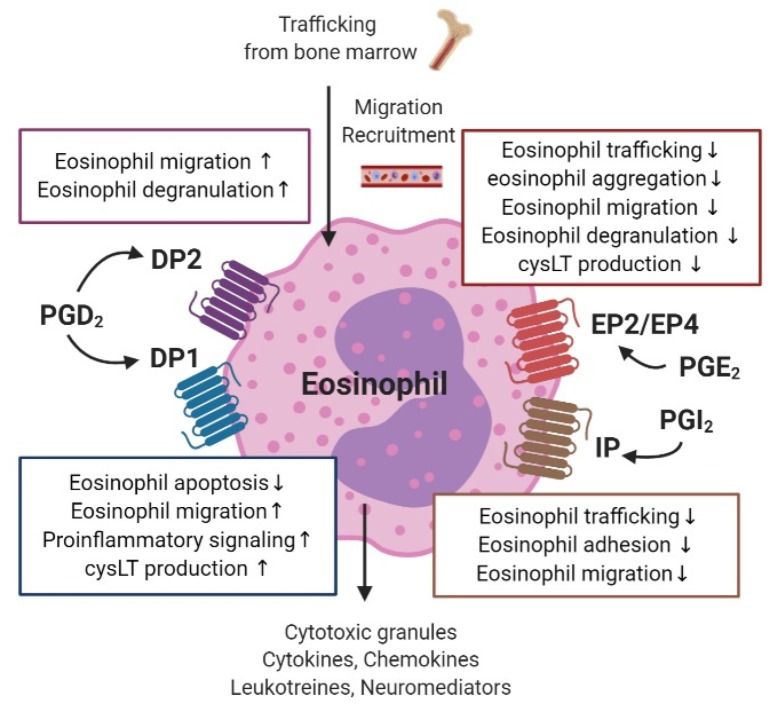
The role of prostaglandins and their receptors in eosinophil function, including bone marrow trafficking, migration, and degranulation.

**Table 1 ijms-21-01851-t001:** Prostaglandins and their specific receptors, downstream signaling, and region of expression.

Ligands	Production	Receptor	Downstream Signaling	Receptor Expression
**PGD_2_**	mast cells, eosinophils, T cells, dendritic cells, macrophages, endothelial cells, platelets, lung parenchyma	DP1	↑cAMP	mucus-secreting goblet cells, nasal serous glands, vascular endothelial cells, T cells, dendritic cells, eosinophils
		DP2 (CRTH2)	↓cAMP, ↑Ca^2+^	T cells, basophils, eosinophils, ILC2
**PGE_2_**	epithelial cells, fibroblasts, macrophage, smooth muscle cells, platelets	EP1	↑Ca^2+^	T cells, dendritic cells, B cells, smooth muscle cells
		EP2	↑cAMP	T cells, dendritic cells, B cells, ILC2, mast cells, basophils, smooth muscle cells
		EP3	↓cAMP	T cells, B cells, dendritic cells, smooth muscle cells
		EP4	↑cAMP	T cells, B cells, dendritic cells, smooth muscle cells
**PGF_2α_**	lung parenchyma, vascular smooth muscle cells, peripheral blood lymphocytes	FP	↑IP3/DAG/Ca^2+^	none
**PGI_2_**	endothelial cells, vascular smooth muscle cells, lung parenchyma	IP	↑cAMP	T cells, dendritic cells, B cells, ILC2, endothelial cells, platelets
**TXA_2_**	platelets, vascular smooth muscle cells, macrophages	TP	↑IP3/DAG/Ca^2+^, ↑↓cAMP	megakaryocytes, monocytes

PG—prostaglandin; cAMP—cyclic adenosine 3′,5′-monophosphate; ILC—innate lymphoid cell; IP—Inositol trisphosphate; DAG—diacylglycerol. An upward arrow (↑) indicates an increased intracellular signaling pathway, and a downward arrow (↓) denotes a decrease in an intracellular signaling pathway.

**Table 2 ijms-21-01851-t002:** Summary of clinical studies on pharmacologic agents targeting PGs in asthma and allergic rhinitis.

Drug and Dose	Indication (Sample Size)	Key Results	Ref.
**DP2 antagonist**			
Fevipiprant (QAW039), oral administration, 500 mg daily for 28 days	Mild to moderate uncontrolled allergic asthma (*n* = 170)	Improvement of lung function in patients with FEV_1_ <70%	[132]
Fevipiprant (QAW039), oral administration, 1–450 mg daily or 2–150 mg twice daily, with inhaled budesonide 200 µg twice a day, for 12 weeks	Allergic asthma uncontrolled by a low-dose inhaled corticosteroid (*n* = 2598)	Total daily dose of 150 mg (150 mg once or 75 mg twice per a day) showed an improvement in forced expiratory volume	[133]
Fevipiprant (QAW039), oral administration, 225 mg twice daily for 12 weeks	Moderate to severe asthma with serum eosinophil ≥ 2% (*n* = 61)	Decreased sputum eosinophil count	[134]
ARRY-502, oral administration, 200 mg twice daily for four weeks	Mild allergic asthma (*n* = 184)	Reduction of FeNO level and decreased serum markers of Th2 inflammation	[135]
AZD1981, oral administration, 100 mg twice daily	Stable asthma withdrawn from inhaled corticosteroid (*n* = 209)	No efficacy on morning peak expiratory flow	[136]
AZD1981, oral administration, 50–1000 mg twice daily, for four weeks, with an inhaled corticosteroid	Asthma uncontrolled by inhaled corticosteroid (*n* = 510)	400 mg group showed improved FEV_1_, significant improvement in questionnaire score and FEV_1_ in atopic subgroup	[136]
OC000549, oral administration, 25 mg daily/200 mg daily/100 mg twice daily, for 12 weeks, with use of short-acting β2 agonist	Mild to moderate asthma (*n* = 460)	Improved FEV_1_ (prominent in eosinophilic subjects), lower incidence of symptom exacerbation and respiratory infection	[137]
OC000549, oral administration, 200 mg twice daily for eight days	Seasonal allergic rhinitis (*n* = 35)	Reduced grass-pollen induced nasal and ocular symptoms	[138]
GB001, oral administration, 30 mg daily for 28 days, with use of low dose inhaled fluticasone propionate	Mild to moderate atopic asthma (*n* = 36)	Improved FEV_1_ (prominent in patients with high FeNO or high blood eosinophil)	[139]
BI671800, oral administration, 50/200/400 mg twice daily for six weeks	Mild to moderate asthma (*n* = 389)	Greater improvement in FEV_1_ compared to moderate doses of fluticasone	[140]
BI671800, oral administration, 400 mg twice daily with inhaled fluticasone (88 µg)	Mild to moderate asthma with inhaled corticosteroid (*n* = 243)	Improvement in FEV_1_ compared to placebo; however, not significantly improved over montelukast	[140]
Setipiprant (ACT-129968), oral administration, 1000 mg twice daily for five days, washout period of three weeks	Stable allergic asthma (*n* = 15)	Reduction in both allergen-induced late asthmatic responses and airway hyper-responsiveness	[141]
Setipiprant (ACT-129968), oral administration, 100/500/1000 mg twice daily or 1000 mg daily for two weeks	Seasonal allergic rhinitis (*n* = 557)	Dose-related improvements in both nasal and ocular symptom scores	[142]
Setipiprant (ACT-129968), oral administration, 1000 mg twice daily for two weeks	Seasonal allergic rhinitis (*n* = 604)	No significant effect on either nasal or ocular symptom scores	[142]
**Dual antagonist for DP2 and TP**			
Ramatroban (BYAu3405), oral administration, 150 mg twice daily for four weeks	Perianal allergic rhinitis (*n* = 10)	Inhibitory effect on allergen challenge-induced nasal mucosal swelling	[143]
Ramatroban (BYAu3405), oral administration, 150 mg twice daily for four weeks	Perianal allergic rhinitis (*n* = 11)	Inhibitory effect on histamine-induced nasal reactivity, decreased eosinophil counts in nasal lavage fluid	[144]
**TP antagonist**			
Seratrodast, oral administration, 80 mg daily for four weeks	Asthma (*n* = 14)	Decreased airway hyper-responsiveness, no definite effect on exhaled nitric oxide and sputum eosinophils	[145]
AA02414, oral administration, 80 mg daily for four months	Asthma (*n* = 31)	Improved symptom score, peak expiratory flow, and bronchial responsiveness to metacholine, decreased activated eosinophil infiltration	[146]
**DP1 antagonist**			
ONO-4053, oral administration, 300 mg daily for two weeks	Seasonal allergic rhinitis (Japanese cedar pollen) (*n* = 200)	Greater improvement in nasal symptoms compared to either placebo or pranlukast	[147]
**PGI_2_ analogue**			
OP-41483, oral administration, 200 µg 4 times daily for four weeks	Stable asthma (*n* = 8)	No direct effect on bronchial responsiveness	[148]

FEV_1_—one second-forced expiratory volume; FeNO—fractional exhaled nitric oxide.

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
