# Peer review of "The Biology of Prostaglandins and Their Role as a Target for Allergic Airway Disease Therapy"

_ijms, 2020, doi:10.3390/ijms21051851_

Round 1
Reviewer 1 Report
To the authors:
The review entitled “The biology of prostaglandins and its role as a target for allergic airway disease therapy” written by the authors: Kijeong Lee, Sang Hag Lee and Tae Hoon Kim, makes a full description of prostaglandins including their biosynthesis and their role in the main immune cells involved in allergic airway diseases such as asthma, allergic rhinitis, nasal polyps and aspirin – exacerbated respiratory disease. The review is highly attractive and sustained by high number of references.
However, there are minor comments / suggestions that I consider the authors should address before acceptance.
Minor comments:
Page 2. After this line: “Among them, groups IIA/V/X of sPLA2, group IVA of cPLA2, and group VI of iPLA2 lead to the generation of lipid mediators.” Please complete the sentence describing which lipid mediators you are referring to? And how they are connected with the PG, which is the aim of the review. Page 2 and forward. The name of “COX-1 and COX-2” in some parts of the manuscript are written in italics and some don’t. Please unify throughout the manuscript. Page 2. After figure 1, “COX-1 is known to not have any function”, please add at the end the word “yet”, because up to date we do not know any function, but you don’t know in the future. Table 1, the footnote does not have the same word size. Page 4, line 1 “decrease in cAMP level”. Please add a line about the consequences of increasing or decreasing the cAMP levels in the cells. Page 4. “mPGES-2 utilizes PGE2 derived from either COX-1 or COX-2 and shows constitutive expression in various cell types.” Please add, which cell types?. Page 4, in the sentence: “TPα activates adenylate cyclase and TPβ inhibits it”, please add a line explaining the importance of adenylate cyclase in the cells. Page 4, the line: “including in the lung”, please delete “in”. Page 5, please add the definition for MUC5AC (a mucin gene expression). Page 5. In the sentence “In the ovalbumin(OVA)-challenged asthma murine model, damaged epithelial cells induced endogenous PGD2 release, and stimulation of DP signaling suppressed eosinophil infiltration, epithelial cell damage, and TNF-α production [35].” I went to the reference, and the 35 does not talks about the ovalbumin(OVA)-challenged asthma murine model, please check. Also check what is written, usually allergic asthma produces eosinophil infiltration and epithelial cell damage, please check carefully. Page 5. After the sentence: “Interestingly, the presence of PGE2 was observed to be effective only at the initiation and not the termination of maturation”, please add “of DC”. Page 5-6. Please change the phrase to “The inhibitory effect of PGD2 administration on DC migration was observed in Balb/c mice exposed to fluorescein isothiocyanate (FITC)-OVA inhalation was observed in a with decreased migration of FITC+ DC to draining lymph nodes.” Page 6, please define “iloprost” and “BMDCs” on the text. Page 7. “Four receptors for PGE2, EP2, and EP4 receptor showed…” Please check, in the manuscript is stated that PGE2 has 4 receptors. Page 7. Please check “Alternaria alternaria”, I think it is “Alternaria alternate” Page 8. “L-PGDS” and “TSLP” please define first in the text before talking about it. Page 9. In the part of “9. Fibroblasts”, please add in nasal polyps (as only information from nasal polyps is given). Page 12. “in vivo studies”, in vivo goes in italics.
Reviewer 2 Report
The review ‘The biology of prostaglandins and its role as a target for allergic airway disease therapy’ by Lee et al. deals with prostaglandins, more precisely their biosynthesis, biological effects on different immune cells and clinical studies on pharmacologic agents modulating prostaglandin signaling.
General comments:
The scope of this review is very broad and this makes it challenging to find a logical structure and stay focused. The authors decided to integrate information of prostaglandin biosynthesis with its effector functions and with preclinical and clinical data. Unfortunately, the authors did not manage to make a consistent review of broad interest and this is mainly due to a pure summary of the literature, without any critical discussions of results or methodological problems. The mix between human and mouse data is partially confusing and sometimes it is not even clear whether the authors describe human or mouse data.
In order to educate the reader and make this review a useful peace of reference I would recommend following:
Decide on the audience you wish to address. The single chapters (biosynthesis, effects on different immune cells, clinical studies) are of diverse interest for the different groups of readers (scientists, clinician scientists, medical doctors, etc.). As an allergologist/immunologist I am mostly interested in a really short, but comprehensive introduction to the biosynthesis of prostaglandins and its receptors followed by a detailed review of the effects on the single immune cells and an overview of current clinical studies in this field. Include the most important information into a table. This can be done as in Table 1 (prostaglandin receptors), but here you should distinguish between ligand and receptor and you could add the main producers of the respective prostaglandin. A table would be also very helpful to summarize the most important clinical studies, here I clearly miss information regarding subject number, key results, etc. This table should include drug, dosing, mechanism of action and references. The clinical studies chapter is very inconsistent. It is only partly about clinical studies dealing with molecules targeting prostaglandin signaling and additionally it describes the role of prostaglandins in AERD and patients with nasal polyps. Again, here you have to decide on your focus and your audience. Please revise the figures carefully. They are very confusing, especially Fig. 2. Maybe you should focus on the single cell types and make a figure for each. Or leave the general overview but then make additionally a graphical summary for each single cell type.
Minor comments:
Add all abbreviations to the list in alphabetical order.Author Response
Please see the attachment.

Reviewer 3 Report
The Authors aimed to address the COX pathway and the potential efficacy of Prostaglandin-targeted therapies.
The review is exhaustive especially in reference to the biology of prostaglandins, which is well known, while there are fewer studies on the potential benefits of drugs directed against prostaglandins or their receptors.
The anti-inflammatory action of NSAIDs, which can paradoxically induce or exacerbate bronchospasm and nasal polyposis, does not seem to improve asthma despite inhibiting the cyclooxygenase pathway.This finding, togheter with the effectiveness of antileukotriene and steroids, strenghtens the relevance of LTs over prostanoids in the pathogenesis of allergic airway diseases: might the Authors stress these aspects too?
Less relevant: although less frequently than first generation NSAIDs, more selective Cox-2 inhibitors can still induce AERD, as well as urticaria, angioedema and anaphylactic symptoms. This consideration should be reported, because saying only that aspirin or COX-1-inhibitor induce AERD could be misleading.
